# Hepatocyte Proteome Alterations Induced by Individual and Combinations of Common Free Fatty Acids

**DOI:** 10.3390/ijms23063356

**Published:** 2022-03-20

**Authors:** Juergen Gindlhuber, Maximilian Schinagl, Laura Liesinger, Barbara Darnhofer, Tamara Tomin, Matthias Schittmayer, Ruth Birner-Gruenberger

**Affiliations:** 1Diagnostic and Research Institute of Pathology, Medical University of Graz, 8010 Graz, Austria; juergen.gindlhuber@medunigraz.at (J.G.); maximilian.schinagl@tuwien.ac.at (M.S.); laura.liesinger@medunigraz.at (L.L.); b.darnhofer@medunigraz.at (B.D.); 2Institute of Chemical Technologies and Analytics, Technische Universität Wien, 1060 Vienna, Austria; tamara.tomin@tuwien.ac.at (T.T.); matthias.schittmayer@tuwien.ac.at (M.S.)

**Keywords:** NAFLD, lipotoxicity, myristic acid, palmitic acid, oleic acid

## Abstract

Non-alcoholic fatty liver disease is a pathology with a hard-to-detect onset and is estimated to be present in a quarter of the adult human population. To improve our understanding of the development of non-alcoholic fatty liver disease, we treated a human hepatoma cell line model, HepG2, with increasing concentrations of common fatty acids, namely myristic, palmitic and oleic acid. To reproduce more physiologically representative conditions, we also included combinations of these fatty acids and monitored the cellular response with an in-depth proteomics approach and imaging techniques. The two saturated fatty acids initially presented a similar phenotype of a dose-dependent decrease in growth rates and impaired lipid droplet formation. Detailed analysis revealed that the drop in the growth rates was due to delayed cell-cycle progression following myristic acid treatment, whereas palmitic acid led to cellular apoptosis. In contrast, oleic acid, as well as saturated fatty acid mixtures with oleic acid, led to a dose-dependent increase in lipid droplet volume without adverse impacts on cell growth. Comparing the effects of harmful single-fatty-acid treatments and the well-tolerated fatty acid mixes on the cellular proteome, we were able to differentiate between fatty-acid-specific cellular responses and likely common lipotoxic denominators.

## 1. Introduction

Maintaining integrity and fluidity of cellular membranes, generating neutral lipids for storage or shuttling fatty acids (FA) into mitochondria for beta oxidation are some of the fundamental constituents of the process called lipid homeostasis [1,2,3]. In this process, the availability of free fatty acids (FFA) is a major driving force, with different tissues tolerating varying FFA levels [4]. Periodical elevation of FFA levels stimulates the formation of lipid droplets (LD), organelles filled with triglycerides and cholesterol esters. Lipid droplets are formed from the endoplasmic reticulum (ER), budding off at designated sites [5]. Permanently elevated FFA levels may lead to ectopic accumulation of lipids and/or enhanced metabolism of FFAs, generating excess amounts of reactive oxygen species [6,7]. Despite variations in the tissue- and cell-type-specific details, this process has been reported to result in increased rates of cell death, inflammation and loss of tissue function [8,9,10,11]. Elevated levels of FFAs are more often than not the result of a Western lifestyle, characterized by a lack of physical activity and unbalanced diet, and comorbidities such as type 2 diabetes and metabolic syndrome [12,13].

In the liver, this pathological process is also more commonly referred to as non-alcoholic fatty liver disease (NAFLD). Initial steps include an elevated amount of liver fat content (≥10%), sometimes accompanied by mild lobular inflammation. This early stage is called non-alcoholic fatty liver (NAFL) and develops with further fat accumulation, progression of the inflammation and potential loss of function into non-alcoholic steatohepatitis (NASH) [14]. Most often, NAFLD reverts on its own and is assumed to occur fleetingly in most individuals of the human population [15]. Long-term elevated FFA levels in plasma can induce a pathological alteration in the hepatocytes’ phenotype known as ballooning cells [16]. These cells can be used as early markers for the progression from NAFLD into NASH, and are thus a potential key to preventing further advancement into fibrosis, cirrhosis and hepatocellular carcinoma (HCC) [17,18]. The number of NAFLD-derived cirrhosis cases is hard to determine, but according to general consensus, the majority of cryptic cirrhosis cases can be attributed to it due to a mostly asymptomatic progression during the early stages [19]. If caught before permanent liver damage has been sustained, current treatment options consist of dietary restriction aiming at a reduction of 5–10% of the current body weight, and eventually, antioxidative supplementation [20,21].

There is a current lack of accurate and easy diagnostic tools or specific treatment options for NAFLD and NASH. In this study, we aimed at improving our understanding of the hepatocellular threshold of FFA toxicity. Thus, we challenged a human hepatoma cell line with different concentrations and combinations of common monounsaturated and saturated FFAs, whilst monitoring phenotypic markers and the composition of the proteome.

## 2. Results

### 2.1. Impact of Fatty Acids on Cell Growth and LD Formation

Growth analysis of HepG2 cells unveiled a dose-dependent decline in growth rates following treatment with either myristic (MA) or palmitic acid (PA) compared to cells receiving only the vehicle (fatty-acid-free bovine serum albumin (BSA)) serving as negative control (Figure 1A). Oleic acid (OA) impacted growth rates only at the highest concentrations tested (i.e., 500 µM). Prevention treatments (PT), MA or PA mixed with OA at ratios of 1:1 (125 µM: 125 µM) or 2:1 (250 µM: 125 µM), respectively, resulted in restoration of the growth rate, completely abolishing the negative impact of the saturated FFAs on cell growth, as shown before in CHO cells [22].

Measuring LDs 24 h after treatment showed that OA induced an increase in average LD volume per cell in an OA concentration-dependent manner (Figure 1B). Saturated FFAs led to an increased number of LDs per cell compared to control, but failed to significantly increase LD volume (Appendix A). PTs showed increased LD volumes overall compared to treatments with any FFA alone. LD volume levels of the FFA co-treatments were comparable to a treatment with 250 µM OA (OA250), which is double the volume we would expect from the OA present in the PTs (125 µM). The only exception is the 1:1 mixture of PA and OA (PO11) treatment, where no increase above LD volume of 125 µM OA-treated cells (OA125) could be observed.

### 2.2. Acid-Specific Proteome Changes

To investigate common and specific cellular responses to different fatty acids, we performed a label-free quantitative (LFQ) proteomic analysis of HepG2 cells treated for 24 h with different concentrations of MA, PA and OA (125, 250, 500 µM) (single FA treatments) or combinations of MA or PA with OA (1:1 (125 µM each) or 2:1 (250 µM: 125 µM) (PTs) (*n* = 3 biological replicates in each group). After removal of contaminants and filtering for three valid values in at least one group, 5322 proteins were reliably identified in our dataset (FDR of 0.01). Single FFA treatments were compared to the BSA control, and statistically significant protein abundance alterations were determined via ANOVA analysis (with S0 set to 0.1, a permutation-based FDR of 0.05 and 250 randomizations).

The PTs were compared to their corresponding saturated FFA concentration using a two-sided Student *t*-test (with S0 set to 0.1, a permutation-based FDR of 0.05 and 250 randomizations). This resulted in a list containing only 38 significantly altered proteins. To create an overview, Venn diagrams including proteins with a fold change of ≥±1.5 were generated and the results were compared (Figure 2A,B). Intriguingly, of the protein expression changes that were found following saturated FA treatment, i.e., 23 after MA and 10 following PA treatment, as well as 6 in the PTs, an increase in PRKDC, a protein associated with sensing DNA damage and required for double-strand break repair, was the only overlap between them (Figure 2C) [23]. Moreover, the two saturated FAs appeared to regulate GRSF1, a regulator of mitochondrial gene expression, in opposite directions [24]. MA led to a dose-dependent increase, whereas protein levels for PA peaked at 125 µM with a 2.9-fold upregulation compared to the control, followed by a dose-dependent reduction at higher concentrations (Figure 2D). Interestingly, HDGF and ROCK2, two proteins involved in cell-cycle progression, exhibited a dose-dependent decline following MA treatment (Figure 3A,B).

String and GO analyses were performed for the 500 µM treatments of single FFAs (using proteins exhibiting a ≥±1.5-fold change compared to control) and for the 2:1 ratio PTs (using proteins exhibiting a ≥±1.5-fold change compared to single FFA treatments) employing the STRING webtool (Appendix A). MA treatment showed upregulated protein clusters connected to mitochondrial biogenesis and beta oxidation and downregulated cytoplasmic translation (Appendix A). These results were similar to PA treatment, except that the latter lacked the cluster with GO terms indicating increased oxidative phosphorylation (Appendix A). OA induced less changes, indicating increased mitochondrial biogenesis and diminished cytosolic translation (Appendix A). PTs containing MA and OA did not only recover cytoplasmic translation but also increased oxidative phosphorylation (Appendix A). PTs containing PA and OA did not show a cluster for cytoplasmic translation but indicated an increase in the mitochondrial electron transport chain (Appendix A).

### 2.3. Fatty-Acid-Specific Effects on Cell Cycle and Viability

To follow up on the MA specific regulation of cell-cycle proteins, we performed a cell cycle and viability flow cytometric analysis of all FFA treatments. Cell-cycle progression was impaired at the highest tested concentration (500 µM) for all FFAs. The only FFA displaying an effect on cell-cycle progression at lower concentrations was MA. The number of cells in the G2 phase were reduced from 22.1% to 13.5% at a concentration of 125 µM (Figure 3C). In addition, propidium iodide and annexin V staining was used to differentiate between necrotic and apoptotic cells. Higher doses of FFAs did not seem to increase the number of apoptotic or necrotic cells in general. Only treatment with PA alone resulted in an increase in apoptotic cells (Figure 3D). Compared to 2.5% in the BSA control, the percentage of apoptotic cells following PA treatment rose to 4.0, 9.0 and 14.6% for concentrations of 125, 250 and 500 µM, respectively. Both MA’s suppression of cell-cycle progression and PA’s induction of apoptosis were abolished by a PT in combination with OA.

### 2.4. Correlation of Potential Cellular Lipotoxicity Protein Biomarkers to Hepatic Cellular Carcinoma Gene Expression and Survival

Significantly altered proteins were manually examined for single fatty acid dose-dependent changes correlating to the observed impact on the proliferation rate (with a major emphasis on protein levels returning to base levels in the PTs) revealing six proteins of interest (Figure 4A). A2M, an interactor of acute phase proteins, and SERPINA3, an acute phase protein, are of great interest on a systemic level but should not influence the cultured hepatocytes [25]. The other four proteins are all involved in transcriptional activity and H2AFY has been recently described as a promising biomarker candidate for HCC patients [26]. Since HCC is a potential terminal outcome of NAFLD, we compared mRNA expression levels of healthy against HCC tissue for DNTTIP2, ZNF326 and LMO7 next to the published HCC biomarker candidate H2AFY in data from the cancer genome atlas liver hepatocellular carcinoma database (*n* = 50; Figure 4B, last accessed 29 November 2021). For DNTTIP2, no difference in expression in HCC compared to healthy liver could be found, although its expression correlated with the survival rate of HCC patients. The 3-year survival rate for ZNF326, on the other hand, was not affected despite being differentially expressed in HCC, thus carrying no prognostic value (Figure 4C). LMO7 was the only protein next to H2AFY found to display changes in protein level following FFA treatment, being differentially expressed at the mRNA level in HCC patients and exhibiting a significant difference in the survival rate (although in an opposite direction).

## 3. Discussion

The two saturated FAs, MA and PA, presented with a strict dose-dependent decline in growth rates until at the highest dose of 500 µM when no further growth could be detected (Figure 1A). At this concentration we were also able to observe a decrease in growth rate for the unsaturated FA, OA. Taking the second phenotypical indicator we measured, the accumulation of LDs, into account, we were able confirm a discrepancy between the LD accumulation following the different FA treatments (Figure 1B) as previously seen by Ricchi and Odoardi et al. [27]. As previously shown for CHO cells by Listenberger et al. [22], lipotoxicity appears to be linked to the capacity to store FAs in TGs. Following this hypothesis, we employed mixed fatty acid treatments, in the following referred to as prevention treatments (PTs), in either an equal ratio of unsaturated FA (OA) to a saturated FA or even in a two to one ratio, with the aim to enhance TG formation. The PTs of MA or PA together with OA had no negative effect on proliferation rates (Figure 1A), which was accompanied by the expected increase in LD volume, with the single exception of PO11 (125 µM PA:125 µM OA). This treatment resulted in an average cellular LD volume of 47.37 µm^3^, which is comparable to the average cellular LD volume of 125 µM OA alone (45.08 µm^3^). Nevertheless, the PO21 (250 µM PA:125 µM OA) treatment doubled the cellular LD volume to the same level observed in the MA:OA co-treatments (Figure 1B). These observations are currently being followed up by lipidomics studies to investigate the incorporation of the different fatty acids into different lipid classes.

Hepatocytes have been shown to react to palmitic acid treatment with induction of apoptosis [28]. Employing an apoptosis/necrosis assay, we corroborated that PA induced apoptosis in HepG2 cells. However, we were surprised that following myristic acid treatment we found a dose-dependent regulation of HDGF and ROCK2, two proteins associated with cell-cycle progression (Figure 3A,B) [29,30]. Our proteomics data were verified by cell-cycle analysis, and despite a noticeable decline in cell-cycle progression by all FFAs at high concentrations, only MA led to a reduction in the number of cells in the G2 phase. To the best of our knowledge, this differential effect on apoptosis and cell-cycle progression between two saturated FFAs within one cell system has not been reported before.

Slowdown of cell growth was also reflected by GO enrichment of the proteomics data, depicting prominent clusters for cytosolic translation found to be downregulated in high FFA conditions. In general, all fatty acids displayed signs of increased mitochondrial biogenesis, which is most likely due to FA-induced activation of peroxisome proliferation-activated receptors [31]. However, enriched clusters indicating enhanced FA oxidation were only found in MA and PTs.

Overall, only 38 out of over 5000 reproducibly quantified proteins were found to be significantly altered in at least one condition after multi-testing correction. Since our aim was to differentiate between the specific cellular response following individual FA treatments and changes that could be considered indicators of lipotoxicity common to all FAs, all proteins were examined for whether changes in their abundance correlated with the observed changes in growth rates induced by single-FA treatments. Proteins were only considered if their levels were altered by single-FA treatments but not by PTs. With this approach, we were able to identify six proteins following this trend (Figure 4A) and two more to be noted, namely PRKDC (Figure 2C) and GRSF1 (Figure 2D). GRSF1 is a regulator of mitochondrial gene expression and the only protein that displayed an opposite dose-dependent expression change comparing treatments with the two different saturated FAs [24,32]. A steady increase in GRSF1 protein abundance in correlation with the treatment could be observed following MA treatment, whilst levels for samples undergoing PA treatment spiked initially at 125 µM and regressed at higher treatments. PRKDC is a sensor for DNA damage and is required for double-strand break repair, playing a role in rRNA biogenesis, mediation of apoptosis and as a housekeeper of hepatic mitochondrial homeostasis [23,33,34,35,36]. Notably, it is the only protein displaying a significant upregulation following treatment with each of the three FFAs, and may be thus linked to general lipid uptake, as reported previously [33].

Two of the proteins identified as potential indicators of lipotoxicity, namely A2M and SERPINA3, are modulators of the innate immune system [37,38,39,40]. Inflammation is one of the key processes in the development and progression of NAFLD, and is linked to the intake and metabolization of FAs [41,42,43]. A2M, which was already found to be upregulated in our previous study focusing on MA treatment, has previously been used as a serum test marker for patients suffering from moderate or advanced hepatic fibrosis. Our results indicate that A2M might be of clinical interest even in earlier stages of pathogenesis [44,45]. Elevated SERPINA3 has been studied for many years due to its connection with neurological diseases such as Alzheimer’s, glioblastoma and multiple sclerosis [46,47]. Recently, it was shown that enhanced SERPINA3 levels result in a poor prognosis for HCC patients due to enhanced transcriptional activity of the hnRNP-K complex [48]. Modulation of hnRNP-K levels can have an oncogenic effect [49]. Interestingly, our results suggest that hepatocytes can affect the innate immune response following FFA treatment, which may thus not solely be dictated by specialised immune cells and their pattern-recognition receptors [50].

It has to be noted that another potential indicator of lipotoxicity we found, H2AFY, has recently been described as a possible biomarker for HCC via a data-mining approach. Our data corroborate that the changes in expression indeed may be due to FA uptake, as hypothesized by the authors [26]. H2AFY, as well as three other proteins of our found potential indicators of lipotoxicity, namely DNTTIP2, ZNF326 and LMO7, are involved in transcription. As HCC is a possible late stage of NAFLD, we further investigated whether they could be used as possible biomarkers employing data of the cancer genome atlas liver hepatocellular carcinoma (LIHC) project. The LHIC project contains gene expression and survival data and is the same dataset which was used in the discovery of H2AFY. Of note, LMO7 expression levels significantly correlated with patients’ survival rates in correspondence with our proteomics results. However, the potential prognostic value of DNTTIP2 and ZNF326 should also not be disregarded, as HCC is only one possible end-stage pathology of NAFLD and the expression data in the LIHC dataset are RNA-based which can differ from protein abundance. Still, a clinical study of NAFLD patients before the stage of advanced fibrosis is needed to evaluate our findings.

In conclusion, our study reveals a regulatory difference between two saturated FFAs within the same cell system which has not been described so far, as well as contextualising proteome alterations following FA treatment. LMO7 was identified as a potential biomarker for HCC progression and A2M and SERPINA3 could play a role not only in modulating the inflammatory process but also in the development and progression of fibrosis and HCC. All findings originating from this study have been summarized in Figure 5. The in vivo validation of our findings remains to be performed.

## 4. Materials and Methods

### 4.1. Experimental Design and Statistical Rationale

Data are presented as mean ± standard deviation error bars of at least 3 biological replicates. Experimental data were analyzed with a two-sided unpaired Students t-test for two groups or ANOVA with Dunnett’s multiple comparisons test for more than two groups. The Mantel–Cox test was employed for comparison of survival curves. Differences were considered significant below a threshold of 0.05 (*p* < 0.05 (*); *p* < 0.01 (**); *p* < 0.001 (***)). If not stated otherwise in the dedicated method section, statistical analyses and generation of graphs were performed with GraphPad Prism (version 8.0).

### 4.2. Reagents

If not stated otherwise, reagents were purchased from Sigma-Aldrich, Vienna, Austria.

### 4.3. Cell Culture

HepG2 cells, obtained from the CellBank Graz, Austria, were cultured at 37 °C and 5% CO_2_ in RPMI-1640 medium supplemented with 10% foetal bovine serum and 584 mg/L glutamine.

### 4.4. BSA Fatty Acid Conjugates and Treatment Regime

To increase solubility and thus improve cellular uptake, fatty acids (FA) were used as bovine serum albumin (BSA) conjugates. The fatty acid: BSA ratio was 6:1. FA-free lyophilized BSA was dissolved in a 150 mM NaCl solution at approximately 37 °C. Myristic (MA; C14:0), palmitic (PA; C16:0) and oleic acid (OA; 18:1 cis-9) sodium salts were slowly heated in 150 mM NaCl solution until the solution turned completely clear. The FA solution was transferred, using a preheated glass pipet to prevent precipitation of the FA, only in tiny amounts, so that the BSA solution never surpassed 40 °C. The conjugate solution was then stirred for 1 h, and volume and pH were adjusted to obtain a 5 mM stock solution. FA-BSA conjugates were added to the growth media 24 h after seeding the cells for 24 h. Treatments with a single FAs are abbreviated with the shorthand form of the FA and the concentration in µM (e.g., MA125 = myristic acid 125 µM). Combinatory treatments are labelled as either MO (mix of MA and OA) or PO (mix of PA and OA) followed by a number indicating the ratio as 11 (equal mix of 125 µM of either fatty acid) or 21 (unequal mix of 250 µM of the saturated FA and 125 µM of OA). The concentrations of fatty acids used in our experiments were determined by their impact on cell growth. The final upper limit, 500 µM, was the lowest concentration at which a negative effect on cell growth following OA treatment could be observed, as well as being the concentration at which growth of saturated FFAs was no longer detectable whilst still being within reported FFA serum levels [51].

### 4.5. Growth Analysis

For cell growth analysis, 80,000 cells were seeded into the wells of a 12-well plate 24 h prior to image acquisition. Growth was analysed through observation of the cells for up to 120 h with a Zeiss (Vienna, Austria) Cell Observer microscope. Seven positions in each well were chosen and images were acquired hourly. Collected images were subjected to an automated confluency analysis and the mean out of all positions in one well was calculated and plotted. An equation was fitted over the plotted data points and analysed for the maximum inclination during their exponential growth phase.

### 4.6. LD Imaging and Volume Analysis

Cells were seeded on KOH-treated cover slips 24 h prior to lipid loading. Further, 24 h after the lipid treatment, cells were incubated with a 1:1000 dilution of BODIPY™ 493/503 (Thermo Fisher Scientific, Vienna, MA, USA) for 10 min at 37 °C to stain intracellular LDs, and fixed for 10 min with 3.7% formaldehyde in PBS followed by two washing steps with PBS. For mounting, VECTASHIELD^®^ antifade mounting medium containing DAPI (Szabo-Scandic, Vienna, Austria) was used. Images were acquired using a Nikon (Vienna, Austria) A1+ confocal laser scanning microscope with a 405 nm diode laser and 450/50 BP filter for DAPI, a 488 nm argon laser with a 525/50 BP filter for BODIPY™ and a 60× CFI Plan Apochromat Lambda oil immersion objective. The settings for x, y and z were chosen to fit the Nyquist criterion. The analysis was performed with FIJI (version 1.51 h). For each image, a maximum intensity projection of the BODIPY™ channel was generated and thresholded using the Kapur–Sahoo–Wong (maximum entropy) method [52]. Calculated values were thereafter used on the whole stack for lipid droplet volume quantification. Based on a nucleus count, an average of lipid droplet volume per cell for each image was calculated.

### 4.7. Cell Cycle and Apoptosis Analysis

A total of 300,000 HepG2 cells were seeded in 6-well plates and grown to approximately 80% confluency. Lipid loading was performed 24 h prior to the analysis on a CytoFLEX LX (Beckman Coulter, Brea, CA, USA). For the cell-cycle analysis, cells were treated with Hoechst 33342 (Thermo Fisher Scientific, Vienna, Austria), followed by washing with PBS and detaching with trypsin. To perform necrosis and apoptosis analysis, cells were first washed with PBS and detached with trypsin. Cells were collected by centrifugation at 220g for 5 min and incubated with Annexin V FITC (BioLegend, San Diego, CA, USA) and propidium iodide for 15 min prior to analysis. In each case, unstained controls and single-stained controls were analysed to determine autofluorescence and to perform necessary compensation. The obtained data were analysed with Cytexpert software (version 2.4).

### 4.8. Proteomics Sample Preparation

In total, 300,000 HepG2 cells were seeded in 6-well plates and grown to approximately 80% confluency prior to lipid loading. After 24 h of treatment, cells were washed 3 times with ice-cold PBS and collected in 300 µL lysis buffer (100 mM Tris pH = 8, 1% sodium dodecyl sulphate (SDS), 10 mM tris (2-carboxyethyl) phosphine, and 40 mM chloroacetamide). A sonication probe at 90% amplitude (counting to 1500 J) was used to lyse the cell suspension, followed by a reduction and alkylation step. A 30-min 3500 g centrifugation step at 4 °C was performed to remove all insoluble debris. Protein content was estimated using the bicinchoninic acid assay (BCA; Thermo Fisher Scientific, Vienna, Austria), after which 100 µg of protein per each sample was precipitated overnight with 3 volumes of acetone. The following day, protein pellets were redissolved in 25% trifluoroethanol (in 100 mM Tris pH = 8.5), diluted to 10% trifluoroethanol with ammonium bicarbonate and digested overnight with trypsin (Thermo Fisher Scientific, Vienna, Austria).

### 4.9. Peptide Pre-Fractionation

We utilised a high-pH reversed-phase C18 column from Waters (Vienna, Austria; Xbridge Peptide BEH 186008979, inner diameter 2.1 mm; length 50 mm; 2.5 µm particle size) for peptide pre-fractionation on an Agilent (Vienna, Austria) 1100 and 1200 series HPLC system. Then, 20 µg of peptides dissolved in solvent A (0.1% triethylamine (*N*,*N*-diethylethanamine), pH 10) was loaded onto the column and a gradient starting with 3% solvent B (100% acetonitrile) rising to 30% B within 30 min was employed with a constant flow rate of 300 µL/min at 50 °C column temperature. The gradient continued to 60% B at 41.3 min, 95% B at 44.6 min and stayed at 95% B for another 2 min followed by a final decrease to 3% B at 54 min. Fractionation collection applied pooling [53]. In summary, 54 fractions (300 µL each) were collected and pooled in an online fashion into 8 fractions using the Agilent (Vienna, Austria) 1200 fraction collector system. Fraction 1 was combined with fractions 9, 17, 25, 33, 41 and 49; fraction 2 was combined with fractions 10, 18, 26, 34, 42 and 50, etc., into Eppendorf low-bind 96-deep-well plates. These deep plates were subsequently dried in a Thermo Fisher Scientific (Vienna, Austria) vacuum concentrator for 24 h at room temperature. Samples were redissolved in 200 µL of a ddH2O: acetonitrile 1:1 (vol:vol) mixture and 0.1% formic acid and further combined to a total of 4 fractions (fraction 1 + 5, 2 + 6, etc.). Finally, samples were transferred to glass vials (Bruckner, Linz, Austria), dried for another 6 h at 30 °C, dissolved in 10 µL 3% acetonitrile + 0.1% formic acid and kept at −20 °C until measurement.

### 4.10. LC-MS/MS Proteomics Measurement

Peptide fractions were analysed by nano-HPLC (Thermo Fisher Scientific, Vienna, Austria, Dionex Ultimate 3000) equipped with an Aurora (IonOpticks, Melbourne, Australia) nanocolumn (C18, 1.6 µm, 250 × 0.075 mm). Separation was carried out at 50 °C at a flow rate of 300 nL/min using the following gradient, where solvent A was 0.1% formic acid in water and solvent B was acetonitrile containing 0.1% formic acid: 0–18 min: 2% B; 18–100 min: 2–25% B; 100–107 min: 25–35% B, 107–108 min: 35–95% B; 108–118 min: 95% B, 118–118.1 min: 95–2% B; 118.1–133 min: 2% B. The maXis II ETD mass spectrometer (Bruker, Vienna, Austria) was operated with the captive source in positive mode with following settings: mass range: 200–2000 m/z, 4 Hz, capillary 1600 V, dry gas flow 3 L/min with 150 °C, nano Booster 0.2 bar, precursor acquisition control top20 (CID).

### 4.11. Proteomics Data Analysis

Data analysis, database searching and quantitation were carried out by MaxQuant (v1.6.1.0) software [54]. The SwissProt human fasta file (downloaded on 4 June 2019, 20467 sequences) containing the most common protein contaminants was used as a database. The false discovery rate (FDR) for peptide, peptide-to-spectrum (PSM), as well as protein matches was set to 1%. Peptide tolerance was ±20 and ±4.5 ppm for the first and main peptide search, respectively. Product mass tolerance was ±0.5 Da. Cysteine carbamidomethylation was set as static and methionine oxidation was set as dynamic modification. Minimum required peptide length was six amino acids and maximum number of allowed tryptic miscleavages was two. Matching between runs was enabled in the retention time window of 3 min and alignment window of 20 min. No intensity threshold for individual spectra was defined. For statistical analysis, Perseus (v1.6.15.0) was employed [55]. Protein quantitation was based on label-free quantitation (LFQ), with a minimum of 2 peptides per protein (unique and razor) as a quantitation requirement. This resulted in a list of 5548 proteins with their corresponding LFQ values. Common contaminants were removed and the matrix was filtered to contain at least 3 valid values in at least one group (every FA concentration was considered a group). This reduced the matrix to 3432 proteins and the remaining missing values were imputed from normal distribution (downshift 1.8, width 0.5). Consequently, ANOVA and Student t-tests were performed with the following criteria: *p*-value of at least 0.05, S0 of 0.1 and permutation-based FDR set to 5% to correct for multi-testing. Venn diagrams were generated using BioVenn webtool [56].

## Figures and Tables

**Figure 1 ijms-23-03356-f001:**
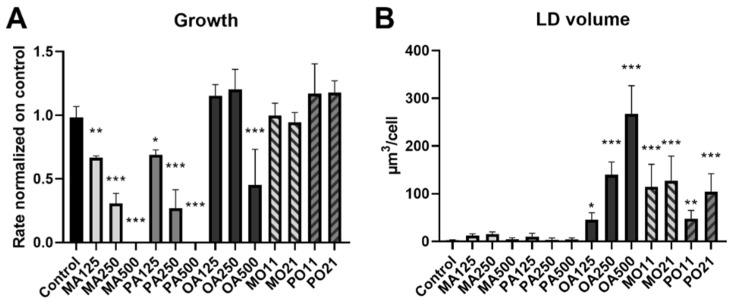
Fatty-acid-specific effects on growth and neutral lipid storage of HepG2 cells. Conditions are labelled as fatty acid and concentration used in µM (MA myristic acid; PA palmitic acid; OA oleic acid) The prevention treatments (PTs) indicate both fatty acids and the ratio (MO myristic and oleic acid; PO palmitic and oleic acid; 11 = 125 µM:125 µM; 21 = 250 µM:125 µM). *p* < 0.05 (*); *p* < 0.01 (**); *p* < 0.001 (***). (**A**) Growth analysis of HepG2 cells treated with FFAs and imaged hourly for 72 h. Confluence increase was plotted over time and the maximum inclination of the resulting function was determined as growth rate readout. Rates presented are a mean out of 7 random positions per well of at least 3 biological replicates. (**B**) LD analysis of HepG2 cells treated with FFAs 24 h prior to imaging. LD volume per cell was calculated from at least 10 independent acquisitions with a total of 1793 cells analyzed.

**Figure 2 ijms-23-03356-f002:**
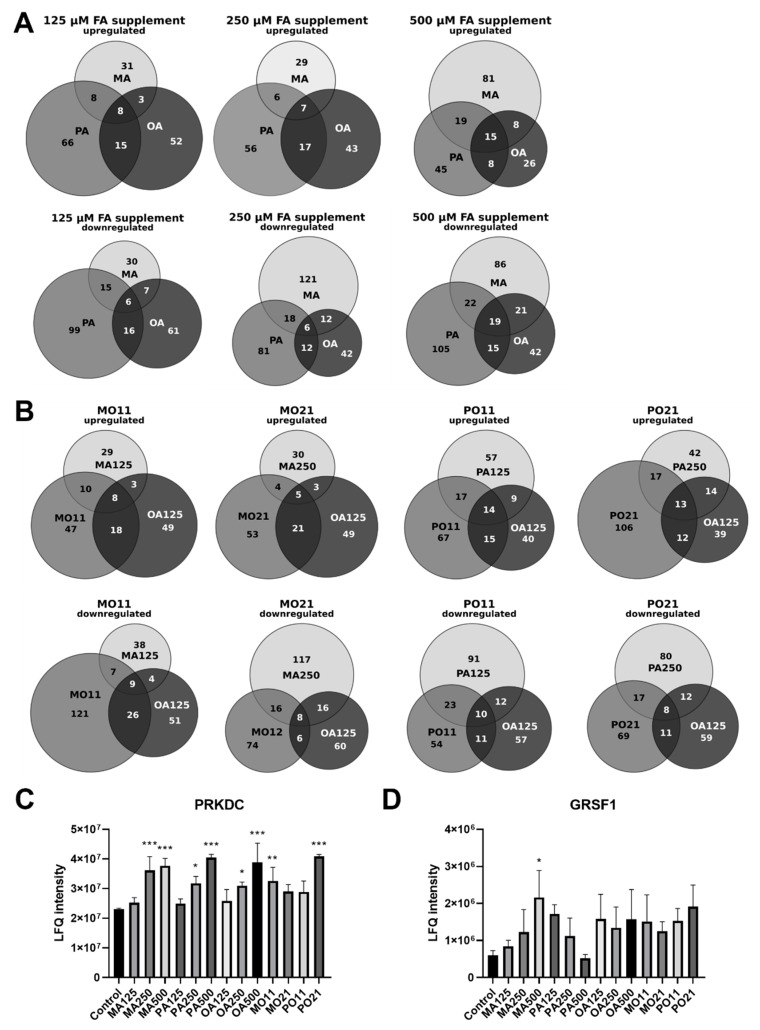
Comparison of fatty-acid-treatment-specific proteome changes. Conditions are labelled as fatty acid and concentration used in µM (MA myristic acid; PA palmitic acid; OA oleic acid). The PTs indicate both fatty acids and the ratio (MO myristic and oleic acid; PO palmitic and oleic acid; 11 = 125:125 µM; 21 = 250:125 µM). (**A**) Venn diagrams displaying the overlap of proteins with a fold change of ≥±1.5 compared to control after single FFA treatments for 24 h; (**B**) Venn diagrams displaying the overlap of proteins with a fold change of ≥±1.5 compared to control after PTs for 24 h; (**C**) PRKDC protein levels; (**D**) GRSF1 protein levels (*p* < 0.05 (*); *p* < 0.01 (**); *p* < 0.001 (***) compared to control).

**Figure 3 ijms-23-03356-f003:**
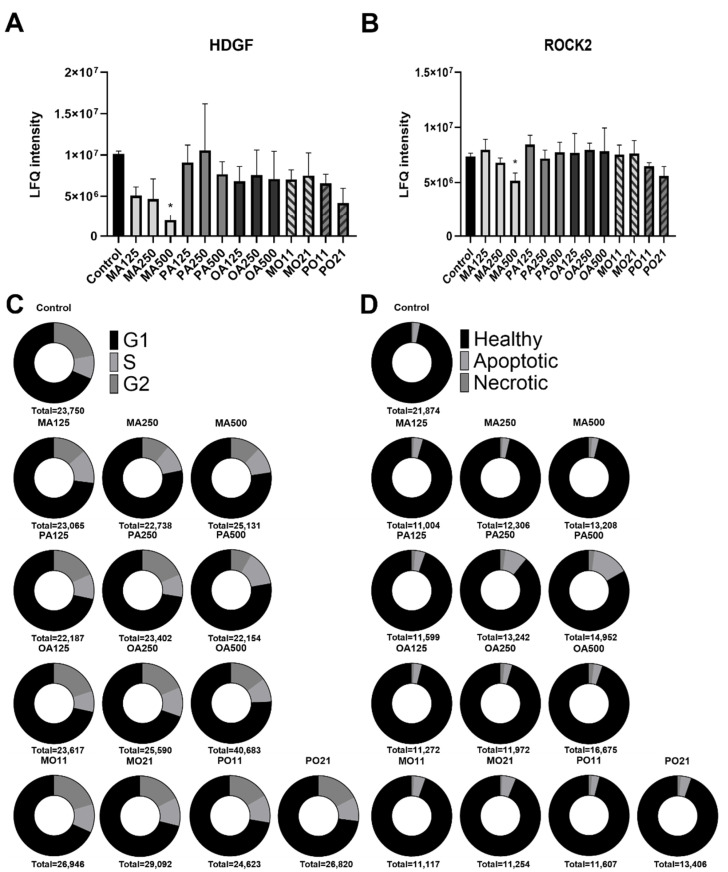
Regulation of proteins involved in cell-cycle progression, cell cycle and cell death in response to fatty acid treatments of HepG2 cells. Conditions are labelled as FFA and concentration used in µM (MA myristic acid; PA palmitic acid; OA oleic acid). The PTs indicate both free fatty acids and the ratio (MO myristic and oleic acid; PO palmitic and oleic acid; 11 = 125 µM: 125 µM; 21 = 250 µM: 125 µM). (**A**,**B**) Protein amounts of HDGF and ROCK2 (*p* < 0.05 (*) compared to control); (**C**). Cell-cycle analysis. Cells were stained (with Hoechst 33342) 24 h after lipid loading. (**D**) Apoptosis/necrosis analysis. Cells were stained with propidium iodide and Annexin V FITC 24 h after lipid loading. Total number of cells counted indicated below each graph in (**C**,**D**).

**Figure 4 ijms-23-03356-f004:**
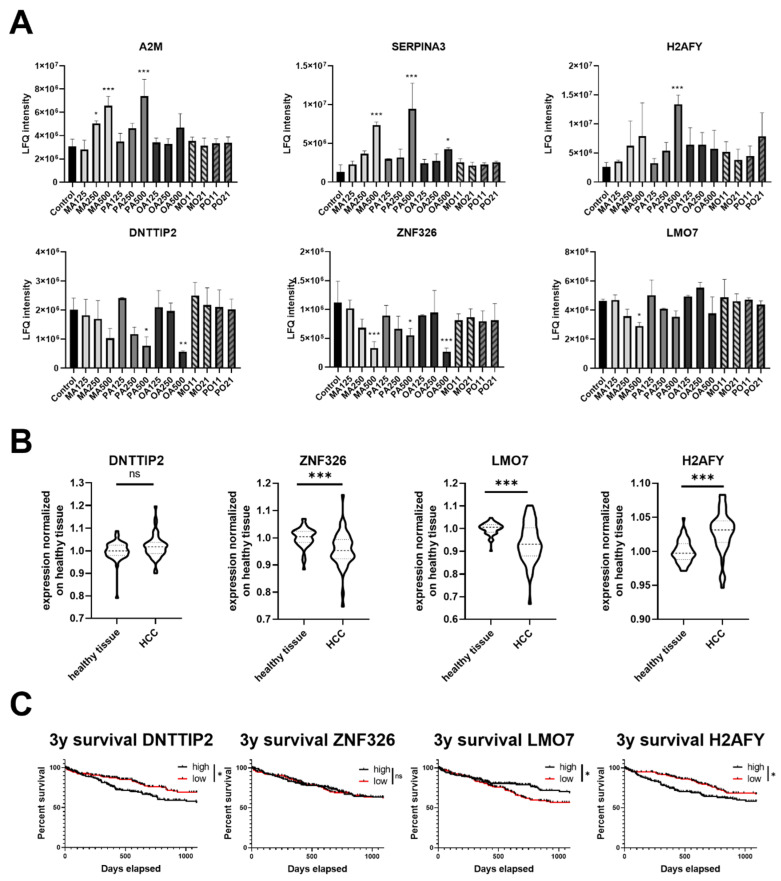
Proteins regulated by saturated fatty acids, their gene expression and correlation with survival in hepatocellular carcinoma. Conditions are labelled as FA and concentration used in µM (MA myristic acid; PA palmitic acid; OA oleic acid) The PTs indicate both fatty acids and the ratio (MO myristic and oleic acid; PO palmitic and oleic acid; 11 125:125 µM; 21 250:125 µM). (**A**) Proteins regulated by saturated fatty acids but not by PTs together with oleic acid; (**B**) mRNA expression levels of healthy tissue compared to HCC samples; (**C**) Kaplan–Meier plots of the 3-year survival rate of HCC patients *p* < 0.05 (*); *p* < 0.01 (**); *p* < 0.001 (***).

**Figure 5 ijms-23-03356-f005:**
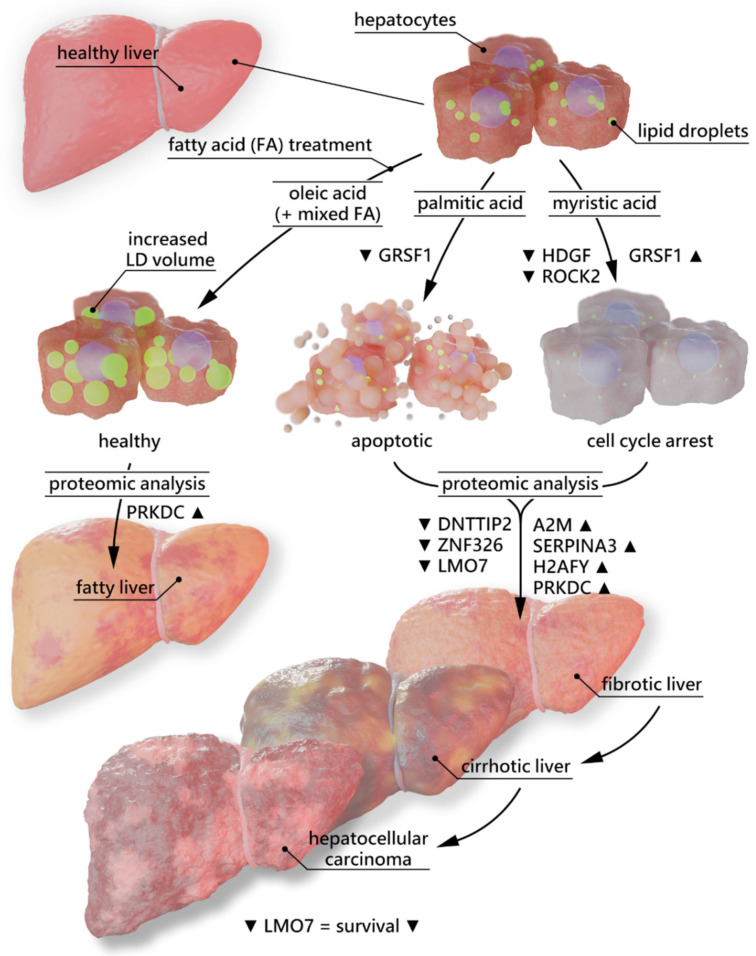
Graphical summary of findings originating from this study.

## Data Availability

The mass spectrometry proteomics have been deposited to the ProteomeXchange Consortium via the PRIDE partner repository with the dataset identifier PXD030764 [57]. Data generated by the TCGA Research Network: https://www.cancer.gov/tcga (last accessed on 29 November 2021) was used in the generation of our results.

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
