# Peer review of "Hepatocyte Proteome Alterations Induced by Individual and Combinations of Common Free Fatty Acids"

_ijms, 2022, doi:10.3390/ijms23063356_

Round 1

Reviewer 1 Report

The manuscript entitled "Hepatocyte Proteome Alterations Induced by Individual and Combinations of Common Free Fatty Acids" corresponds to an interesting study. The methodology used is appropriate and consistent with the purpose of the study. The results are sufficient and support the discussion and conclusion. However, I have the following comments.

I. Major Comments:
1. In the introduction it is necessary to include a brief paragraph regarding the diet and the development of NAFLD. Especially those dietary aspects related to the high intake of fatty acids, for example palmitic.

2. Lipotoxicity and the development of NAFLD are directly linked to oxidative stress. Briefly refer to this point in the introduction.

3. The discussion is good and is based on the results. However, I suggest improving some points.
3.1. Fatty acids (PA) can increase the inflammatory response (activation of NF-kB).
J Nutr Biochem. 2019;63:35-43. doi: 10.1016/j.jnutbio.2018.09.012.

3.2. It is important to discuss some molecular aspects related to oxidative stress, lipotoxicity and inflammatory response.

3.3. Fatty acids regulate the activity of some transcription factors (SREBP-1c and PPAR-alpha), factors that influence mitochondrial function and alter fatty acid oxidation.

4. In the discussion, it is important to briefly link these results to the development of NAFLD. For example; Characteristics of the diet of people with hepatic steatosis, progression to NASH, and possible clinical interventions.

5. What potential clinical projection would this study have? Would it be possible to propose a new study to see possible prevention or reversal interventions? Briefly discuss.

6. It would be very good for the manuscript to include in the discussion a figure that summarizes the observed results.

II. Minor comments:
1. Improve the writing of the objective of the study.
2. Figure 1. It is possible to include Figure 1C as an independent figure. It would be very good for the manuscript (better understanding for readers, especially students).

Author Response

Esteemed Reviewer 1

Thank you very much for your time and positive criticism. We addressed your comments point by point below.

Major Comments

1+2. The impact of the diet and ROS stress has now been stressed in the introduction as well as in the discussion.

3. The interaction between the immune response and oxidative stress has been clarified and made more prominent in the discussion.

4. We made an effort to better contextualize the findings of this study within the development of NAFLD and progression to NASH fibrosis and HCC.

5. Possible clinical significance was added to the discussion.

6. The newly added graphical abstract and restructured discussion including a new summary figure should improve understanding and result in an easier reading experience.

The minor comments were all considered. Fig. 1C is now expanded with all conditions and can be found as supplemental figure 1.

Reviewer 2 Report

The manuscript investigates the proteome changes occurring in human HepG2 cell lines treated with saturated fatty acids (myristic and palmitic acid) alone, or in combination with the monounsaturated oleic acid.

The research design and the manuscript present critical issues.

  • First of all, in the result section two Figures 2 are present: this causes an incongruity between the figure description in the text and the Figures, and renders reading the manuscript very difficult.

A careful revision of the manuscript must to be done.

  • The experiments have been carried out using very high fatty acid concentrations. Moreover, the authors affirm that these concentrations have been determined empirically. The aspect of the experimental design has to better discussed because it has been previously reported in literature that HepG2 cell viability is affected by saturated fatty acids at lower concentrations. Moreover, since the protective effect of oleic acid against satuarted fatty acids has been already reported, the action of polyunsaturated fatty acid should be investigated to complete the study on the effets of unsaturated fatty acids.
  • The authors define the co-treatment with myristic acid plus oleic acid or palmitic acid plus oleic acid “a rescue treatment” able to restore the proliferation rate at the level of control cells.

This statement is conceptually incorrect because the two fatty acids (saturated plus unsaturated) have been administered simultaneously. For this reason, it is correct to speck of prevention effect, not of reversion. This aspect has to be modified in all the manuscript and in the Figure Legends

  • In the Abstract, the authors affirm that “The two saturated fatty acids presented a similar phenotype of a dose dependent decrease in proliferation rates….We demonstrated that the drop in the proliferation rates was due to delayed cell cycle progression following myristic acid treatment whereas palmitic acid led to cellular apoptosis.” This statement has to be changed, because it is incorrect to affirm that a reduction of proliferation is due to cellular apoptosis. In this light, also in the Discussion the comment on this part of the results must to be revised.
  • In some paragraphs of the Results, references to other publications are present. Usually, in this section only the comments to the results of the current work have to be included. Moreover, the specific data referred to LD volume present in the Discussion should to be described in the Results.
  • In the Figure Legends, no comments to the results and description of the molecule actions must be present.

Minor criticisms

  • 4.3 Cell culture. There are not antibiotics in the culture medium?
  • The number of seeded cells has to be specified for all the treatments and expressed with reference to the unit surface area, mainly because the type of multiwells differs in the different experiments.

Author Response

Esteemed Reviewer 2

Thank you very much for your time and positive criticism. Please find our point by point response below.

There has been an issue with the sequential numbering in word and we beg your pardon. The issue has been addressed by overwriting words automated input.

Regarding the major comments.

As mentioned at the beginning of the discussion the treatment concentrations used are well within reported serum levels (Abdelmagid et al 2015 doi: 10.1371/journal.pone.0116195). This study focussed on lean and healthy (mean BMI 22.8 ±3.4) young individuals and measurements were taken randomly, not postprandial. As the means of these measurements are beyond the 1 mM mark our treatments cannot be considered very high. The term empirically has been removed and the sentence rephrased. The impact on viability can vary a lot depending on the setup especially with regard to treatment time as well as absence/ presence of FBS in the media. Viability was examined and reported in this manuscript for specific treatment concentrations and times.

The protective effect for oleic acid is used as a tool to differentiate between results which occur due to the process of lipid loading and those which are only present when adverse effects to the proliferation rates could be observed. We completely agree with the reviewer that the term “prevention treatment” is a better fit and adapted it into the manuscript.

The abstract has been adapted to no longer be ambiguous. The term proliferation was exchanged by cell growth. As cell growth was measured indirectly via confluency a distinction cannot be made from this data alone and clarification if a lack of mitotic activity or an increased number of cells dying is needed.

The results and discussion sections were partly reorganized and a new summary figure was added to improve the structure and readability. Results were removed from the Figure legends.

Minor comments

No antibiotics were present in the culture media. Numbers of cells used have been specified for the different well sizes.

Reviewer 3 Report

The paper “Hepatocyte proteome alterations induced by individual and combinations of common free fatty acids” reports on the effects of specific saturated and unsaturated fatty acids on a human hepatoma cell line model, with the aim to investigate their influence in the development of NAFLD. The results are interesting as data are given about the effects of fatty acids administration on different functions of the cells. However, the text needs to be revised as detailed below, and thus requires a major revision.

In general, the numerous and different kinds of data presented are uneasy to track by the reader. The organization of the paper needs to be improved, as to both the presentation of the Figures, and as to some details about the procedures used, as given below.  

Also, a table reporting the kinds of analyses which have been performed, the related abbreviation used along the text, and the meaning of the test performed will make easier for the reader to appreciate the meaning of results.

Major remarks

Discussion- page 5- The first sentence is to be moved to M&M, since it justifies the choice of dose treatment; also, the term “empirically” is unproper, since the choice was based on proliferation rate assay, is it?

Some of the following sentences report results here obtained and references, anyway without any actual discussion. For example, it is not clarified if references is provided to compare data from the paper and from literature as to a possible agreement, or simply by using the reference as to the ability of the type of analysis performed to detect the target of interest.

Minor remarks

The numbering of the images has to be checked and corrected. Figure 1 is not recalled along the text, there are two Figures n.2, while they seem to have been recalled correctly in the text, and an effort should be done to present all the Figures in the results section.

Page 9- please specify the target for BODIPY™ 493/503, and revise the legend of Figure 1C, by stating which dye, and related color, identify nuclei and lipids.

Page 10, 4.7. Flow cytometry- This title should be improved, by adding the biological parameter (cell cycle), to conform with other titles.

Author Response

Esteemed reviewer 3

Thank you very much for your time and positive criticism.

Major remarks

This sentence was moved to the material and methods section, the word empirically was removed and the sentence was rephrased for the sake of clarity.

The whole discussion section was rephrased and a new summary figure of our findings (instead of the suggested table) was added to provide more context for the results and facilitate better understanding. We made an effort to improve overall clarity.

Minor remarks

There has been an issue with the sequential numbering in word and we beg your pardon. The issue has been addressed by overwriting words automated input.

Also all other minor remarks were addressed and the manuscript adapted accordingly.

Round 2

Reviewer 1 Report

Authors made all changes suggested. My concerns are well addressed. Therefore, the manuscript can be accepted in the present form.

It is an interesting manuscript.
I was able to identify some changes in the manuscript. But it is necessary that the authors identify with some color (example: yellow) the changes made in the manuscript. Only then, I will be able to evaluate all the changes made by the authors.

Reviewer 2 Report

The authors addressed the major comments and modified the manuscript following the reviewer's suggestions.

In this form the manuscript is suitable for publication after minor changes (for example, the abbreviation "prevention treatment (PTs)" is repeated several times in the text).

Reviewer 3 Report

The paper “Hepatocyte proteome alterations induced by individual and combinations of common free fatty acids” have duly revised after having carefully considered the comments of the Reviewer. Particularly appreciated the addition of Figure 5, clearly guiding the reader along the sequence of the findings from the study with the representation of a schematic, graphical summary. Therefore, the paper is no suitable for publication in “IJMS”.